# Aflatoxin B_1_ Increases Soluble Epoxide Hydrolase in the Brain and Induces Neuroinflammation and Dopaminergic Neurotoxicity

**DOI:** 10.3390/ijms24129938

**Published:** 2023-06-09

**Authors:** Weicang Wang, Yuxin Wang, Karen M. Wagner, Ruth Diana Lee, Sung Hee Hwang, Christophe Morisseau, Heike Wulff, Bruce D. Hammock

**Affiliations:** 1Department of Entomology and Nematology, and UC Davis Comprehensive Cancer Center, University of California, Davis, CA 95616, USA; wcwwang@ucdavis.edu (W.W.); yxwa@ucdavis.edu (Y.W.); kmwagner@ucdavis.edu (K.M.W.); shhwang@ucdavis.edu (S.H.H.); chmorisseau@ucdavis.edu (C.M.); 2Department of Pharmacology, University of California Davis, Davis, CA 95616, USA; rdlee@ucdavis.edu (R.D.L.); hwulff@ucdavis.edu (H.W.)

**Keywords:** neuroinflammation, dopaminergic neurotoxicity, aflatoxin B_1_, soluble epoxide hydrolase

## Abstract

Parkinson’s disease (PD) is an increasingly common neurodegenerative movement disorder with contributing factors that are still largely unexplored and currently no effective intervention strategy. Epidemiological and pre-clinical studies support the close association between environmental toxicant exposure and PD incidence. Aflatoxin B_1_ (AFB_1_), a hazardous mycotoxin commonly present in food and environment, is alarmingly high in many areas of the world. Previous evidence suggests that chronic exposure to AFB_1_ leads to neurological disorders as well as cancer. However, whether and how aflatoxin B_1_ contributes to the pathogenesis of PD is poorly understood. Here, oral exposure to AFB_1_ is shown to induce neuroinflammation, trigger the α-synuclein pathology, and cause dopaminergic neurotoxicity. This was accompanied by the increased expression and enzymatic activity of soluble epoxide hydrolase (sEH) in the mouse brain. Importantly, genetic deletion or pharmacological inhibition of sEH alleviated the AFB_1_-induced neuroinflammation by reducing microglia activation and suppressing pro-inflammatory factors in the brain. Furthermore, blocking the action of sEH attenuated dopaminergic neuron dysfunction caused by AFB_1_ in vivo and in vitro. Together, our findings suggest a contributing role of AFB_1_ to PD etiology and highlight sEH as a potential pharmacological target for alleviating PD-related neuronal disorders caused by AFB_1_ exposure.

## 1. Introduction

Parkinson’s disease (PD) is a neurodegenerative disorder affecting millions of people with rising global prevalence [1,2]. Common features of PD are the appearance of α-synuclein-containing Lewy bodies, loss of dopaminergic neurons, and deficiency of dopamine, which leads to both motor and non-motor symptoms [3]. The etiology of PD remains unclear, while accumulating evidence supports that environmental risk factors act as major contributors to PD [4,5]. Indeed, prolonged exposure to environmental pollutants, including various chemical and biological toxins, has been shown to cause PD not only by triggering pro-inflammatory responses and oxidative stress, but also by promoting Lewy body formation and neuronal dysfunction in the brain [6,7]. It is possible that there are multiple contributing factors to sensitivity to Parkinson’s disease. There currently are no effective therapies that slow down or stop the progression of PD [1]. Given the serious individual and financial burdens caused by the increasing incidence of PD, there is an urgent need to better understand the neurotoxic effect of environmental risk factors and establish target-based strategies for preventing and attenuating the course of PD.

Aflatoxin B_1_ (AFB_1_), a mycotoxin produced from *Aspergillus flavus* and *Aspergillus parasiticus*, is commonly found in a wide range of food commodities, including maize, peanuts, wheat, and peppers [8]. It is estimated that 4.5 billion people are chronically exposed to AFB_1_ via contaminated food worldwide [9]. In addition to its genotoxic, mutagenic, and immunotoxic effects, emerging evidence supports the accumulation of AFB_1_ and subsequent neurotoxicity in the brain [10,11]. Notably, AFB_1_ was detected in brain samples of children who were exposed to contaminated food in Nigeria [12]. Animal studies showed that exposure to AFB_1_ disrupts cognitive function, impairs spatial memory, and promotes anxious depressive behaviors in mice [13,14]. Prenatal exposure to AFB_1_ in rats leads to deficits in both motor coordination and learning ability in offspring, suggesting that AFB_1_ affects neuronal development and function [15]. In addition, AFB_1_ triggers cytotoxicity, induces pro-inflammatory reactions, and elicits oxidative damage in neurons, microglial, and astrocytes in vitro [16,17,18]. However, the mechanism by which AFB_1_ induces neuroinflammation and affects the function of dopaminergic neurons in vivo remains to be elucidated.

Oxylipins, the lipid mediators that are derived from polyunsaturated fatty acid (PUFA) metabolism, play a vital role in controlling inflammation and the pathophysiology of neurological diseases [19,20,21]. Soluble epoxide hydrolase (sEH) is a crucial oxylipin-metabolizing enzyme, which converts the inflammation-resolving epoxy fatty acids into pro-inflammatory diols [22]. Recent evidence showed that sEH is actively involved in the pathogenesis of a series of neurological disorders, including Alzheimer’s disease, depression, and autism [23,24,25]. In particular, sEH contributes to the pathogenesis of PD induced by environmental toxins. Indeed, in the 1-methyl-4-phenyl-1,2,3,6-tetrahydropyridine (MPTP)-induced mouse model of PD, genetic ablation, or pharmacological inhibition of sEH, attenuated dopaminergic neuron loss, reduced oxidative stress, and improved motor performance in the MPTP-treated mice [26,27]. Inhibition of sEH also alleviated paraquat- or rotenone-induced neuroinflammation and dopaminergic neurotoxicity [26,28]. However, whether sEH is involved in the pathogenesis of AFB_1_-induced neurotoxicity is still unknown.

To this end, the effect of AFB_1_ on neuroinflammation and dopaminergic neurotoxicity was investigated in a C57BL/6J mouse model. Using sEH genetic knockout mice (*Ephx2*^−/−^ mice) and a potent sEH inhibitor, EC5026, we tested the functional role of sEH in AFB_1_-associated pro-inflammatory response and neuronal dysfunction.

## 2. Results

### 2.1. AFB_1_ Increases Neuroinflammation

We orally exposed 8-week-old male C57BL/6J mice to AFB_1_ (1500 ppb) in drinking water for 21 days. The AFB_1_ exposure dose was selected as comparable to the contamination levels in food and feed materials that people could be exposed to [29,30,31,32]. Microglia are resident immune cells in the brain and are crucial for immune surveillance and regulating inflammation [33]. AFB_1_ exposure increased the gene expression and protein levels of Iba-1, indicating microglial cell activation in the brain (Figure 1A,B). In line with microglia activation, the qRT-PCR analysis also showed increased expression of pro-inflammatory cytokines *Il-1β*, *Mcp-1*, *Csf-2*, and *Cxcl-10* in the brain (Figure 1C). Furthermore, the level of the inhibitor of NF-κB alpha (IκBα) was decreased after 21 days of AFB_1_ exposure, suggesting the activation of the NF-κB pathway in the brain (Figure 1D). Together, these results support the premise that AFB_1_ induces microglia activation and inflammation in the brain.

### 2.2. AFB_1_ Stimulates α-Synuclein Upregulation and Causes Dopaminergic Neurotoxicity

The accumulation of α-synuclein and dopaminergic neuron dysfunction are two hallmarks and driving factors in PD pathology [3]. The qRT-PCR analysis showed that AFB_1_ increased the expression of α-synuclein in the brain, indicating the involvement of AFB_1_ in α-synuclein pathology (Figure 2A). Moreover, AFB_1_ exposure suppressed the expression of the dopaminergic neuron marker tyrosine hydroxylase (TH) and decreased the level of dopamine in the brain, suggesting dopaminergic neuron dysfunction (Figure 2B,C). Consistent with these in vivo findings, treatment with AFB_1_ dose-dependently reduced the cell viability of N27 rat dopaminergic neuronal cells in vitro (Appendix A). Together, these results support that AFB_1_ leads to dopaminergic neurotoxicity.

### 2.3. AFB_1_ Induces Neuroinflammation in an sEH-Dependent Manner

sEH contributes to environmental toxin-induced parkinsonism in mice [26]. Here, AFB_1_ exposure increased the gene expression and enzymatic activity of sEH in the brain, suggesting its potential involvement in AFB_1_ toxicity (Figure 3A,B). To determine the functional role of sEH in AFB_1_-induced neuroinflammation, 8-week-old male wild-type (WT) or sEH genetic knockout (*Ephx2*^−/−^) mice were treated with AFB_1_ for 21 days. Moreover, the small-molecule transition state inhibitor EC5026 was applied as a pharmacological intervention approach to block the activity of sEH in mice. sEH deficiency or inhibition attenuated AFB_1_-induced neuroinflammation by suppressing the accumulation of Iba-1^+^ microglia cells, decreasing the levels of pro-inflammatory cytokines (*Il-1β*, *Mcp-1*, *Csf-2*, and *Cxcl-10*), and suppressing the activation of the NF-κB pathway in the brain (Figure 3C–G). Together, these results support the involvement of sEH in mediating the pro-inflammatory effects of AFB_1_ in the brain.

### 2.4. sEH Deficiency or Inhibition Alleviates AFB_1_-Induced α-Synuclein Pathology and Dopaminergic Neurotoxicity

Having demonstrated a functional role of sEH in modulating AFB_1_-associated neuroinflammation, we next tested whether sEH is also involved in mediating the AFB_1_-induced α-synuclein pathology and dopaminergic neurotoxicity in the brain. Genetic knockout or pharmacological inhibition of sEH reduced the AFB_1_-induced α-synuclein upregulation in the brain (Figure 4A,B). Phosphorylation of α-synuclein, which controls protein aggregation and Lewy body formation [34,35], was also decreased in *Ephx2*^−/−^ or EC5026-treated mice (Figure 4B). Moreover, sEH deficiency or inhibition prevented the loss of the dopaminergic neuron marker TH and attenuated the reduction in dopamine in the brain under AFB_1_ exposure, pointing to the alleviated dopaminergic neuron damage after sEH inhibition (Figure 4C,D). In N27 rat dopaminergic neuronal cells, treatment with EC5026 suppressed the AFB_1_-induced oxidative marker *Inos*, as well as pro-inflammatory cytokines *Tnf-α* and *Cxcl-10*. (Appendix A). Together, these results support the vital role of sEH in modulating the α-synuclein pathology and dopaminergic neurotoxicity caused by AFB_1_ in the brain.

## 3. Discussion

The incidence and prevalence of PD have risen rapidly worldwide [1,2]. Epidemiological and animal studies reported a higher risk of onset and progression of PD due to environmental toxicant exposure [4,5]. However, whether AFB_1_, a widespread and dangerous foodborne mycotoxin, is involved in the pathogenesis of PD is poorly understood. Here, the results suggest that exposure to AFB_1_ could contribute to PD by stimulating neuroinflammation, triggering α-synuclein pathology, and impairing the function of dopaminergic neurons in vivo. Moreover, both the expression and enzymatic activity of sEH are increased in the brains of AFB_1_-exposed mice. Blocking of sEH, genetically or pharmacologically, attenuates the AFB_1_-induced neuroinflammation, α-synuclein accumulation, and dopaminergic neurotoxicity in mice. This is consistent with earlier evidence that an increase in sEH expression or activity is both a marker and cause of inflammation in the brain. Altogether, these results support that AFB_1_ can induce neuroinflammation and PD-like pathology through a mechanism that involves the increased level of sEH.

Neuroinflammation has been implicated in the pathogenesis of a variety of neurodegenerative diseases, including PD [36]. Here in mice, AFB_1_ exposure induces microglia activation and leads to activation of the NF-κB pathway, and up-regulation of the expression of pro-inflammatory cytokines, notably *Il-1β*, *Mcp-1*, *Csf-2*, and *Cxcl-10*, in the brains of AFB_1_-exposed mice. In line with these findings, previous studies showed that AFB_1_ stimulated the production of pro-inflammatory cytokines and induced the activation of NF-κB in human or mouse microglia cell lines [16,17]. Promisingly, sEH deficiency or inhibition suppressed the AFB_1_-induced Iba-1^+^ microglia accumulation and pro-inflammatory cytokine up-regulation in the brain, indicating an important role for sEH in mediating neuroinflammation following AFB_1_ exposure. These results agree with previous reports showing that blocking the action of sEH attenuated neuroinflammation by decreasing the abundance of Iba-1^+^ microglia and levels of pro-inflammatory regulators in the brains of 5xFAD- or LPS-treated mice [37]. In vitro, the treatment with an sEH inhibitor reduced the expression of pro-inflammatory cytokines in LPS-treated microglia or rotenone-treated N27 dopaminergic cells [28,37]. Together, the oral intake of AFB_1_ increased neuroinflammation via increased sEH, which could further contribute to neurological changes toward PD in the brain.

Dopaminergic neuron damage is a core manifestation of PD [38]. Here, AFB_1_ suppressed the viability of N27 dopaminergic neurons in vitro and caused dopaminergic neuron dysfunction in mice. Similar effects have been also described in models of dopaminergic dysfunction in rats or *C. elegans* where AFB_1_ causes dopaminergic neurodegeneration and suppresses the level of dopamine in the brain [39,40], suggesting that AFB_1_ causes dopaminergic neurotoxicity in multiple animal models. Exposure to AFB_1_ also causes nerve fiber depletion [41]. In this study, inhibiting the function of sEH attenuated AFB_1_-induced dopaminergic neurotoxicity by restoring the expression of dopaminergic neuron marker TH and the level of dopamine in the brain. In agreement with these findings, previous studies showed that sEH inhibition reduced MPTP- or paraquat-induced TH^+^ dopaminergic neuron loss [26]. Moreover, treatment with an sEH inhibitor restored dopamine levels in MPTP-treated mice [42] or rotenone-treated *Drosophila melanogaster* [28], suggesting that sEH contributes to environmental toxin-induced dopaminergic neurotoxicity. A limitation of the current study is it may not reflect the sex difference in AFB_1_-caused dopaminergic neurotoxicity. Further studies are needed to better characterize the pro-PD effects of AFB_1_ on both male and female mice. Another limitation of this study is that we only analyzed dopaminergic neuronal damage in frontal brain regions, mainly the striatum; however, whether AFB_1_ also causes toxicity to dopaminergic neurons in the substantia nigra (SN) is still unclear. Further studies are needed to determine dopaminergic neuronal dysfunction and inflammation and to evaluate the functional role of sEH in mediating neuroinflammation and neurotoxicity in the SN under AFB_1_ exposure.

In addition to its direct toxic effect on dopaminergic neurons, AFB_1_ also increased the level and phosphorylation of α-synuclein in the mouse brain. It is well-known that α-synuclein acts as the primary structural component of Lewy-bodies and contributes to progressive neuronal damage during PD development [43,44]. In particular, Ser129 phosphorylation of α-synuclein has been shown to control the aggregation process and has been linked to neurotoxic effects in the pathogenesis of PD [34,35]. Moreover, α-synuclein stimulates the inflammation response in microglia, which in turn boosts α-synuclein pathology and dopaminergic neuronal death [45,46]. A recent study also highlights that pathogenic gut microbes contribute to PD development by inducing α-synuclein aggregation [47]. Thus, targeting synuclein, especially its phosphorylation, is an attractive approach for preventing Lewy body formation and PD development. Here, inhibiting the action of sEH helps to attenuate AFB_1_-stimulated α-synuclein upregulation and phosphorylation in the brain. Similarly, treatment with an sEH inhibitor has been shown to reduce phosphorylated α-synuclein in the brains of MPTP-treated mice [42]. Further studies are needed to determine whether AFB_1_ has a direct interaction with a-synuclein, whether it affects aggregation/fibril formation in vitro and in vivo, and how sEH exactly is involved in such aggregation processes.

In the current study, both the expression and enzymatic activity of sEH were found to be increased in the brains of AFB_1_-exposed mice. Additionally, sEH has been reported to be upregulated in the brains of MPTP- or glyphosate-treated mice [26,42,48], suggesting the broad involvement of sEH in neurological disorders induced by environmental factors. In humans, the level of sEH is increased in the brains of Lewy body dementia patients [42]. sEH mainly exhibits its pro-inflammatory role by converting epoxy fatty acids, especially epoxyeicosatrienoic acids (EETs), into responding diols. Indeed, previous studies showed that 14,15-EET help to restore the MPTP-disrupted dopaminergic neurons and improve rotarod performance in mice [26]. In addition, the treatment of 14,15-EET protects the dopaminergic neuronal N27 from oxidative damage [49]. Given the neuron protective effects of 14,15-EET, further studies investigating the functions of 14,15-EET in AFB_1_ neurotoxicity models are warranted to further consolidate the molecular mechanism by which the inhibition of sEH by stabilizing 14,15-EET achieves its neuroprotective effects.

To facilitate translation to humans, AFB_1_-exposed mice were treated with the sEH inhibitor EC5026. We found that the treatment of EC5026 effectively attenuated neuroinflammation and dopaminergic neurotoxicity. In N27 dopaminergic cells, the treatment of EC5026 blocked the AFB_1_-upregulated pro-inflammatory factors *Inos*, *Tnf-α* and *Cxcl-10* in vitro. These findings are in agreement with previous studies showing that EC5026 reduces LPS-induced neuroinflammation in mice [37] and alleviated neuropathic pain in a chronic constriction injury rat model [50]. All these findings are of translational relevance, since EC5026 is currently in clinical trials for neuropathic pain [50]. Further studies are needed to explore the efficacy and mechanism of EC5026 in treating AFB_1_-caused neurological and behavioral disorders in classic PD models.

In conclusion, AFB_1_ oral exposure promotes PD pathogenesis by enhancing neuroinflammation and disrupting the dopaminergic neuron function in mice. More importantly, sEH acts as a critical cellular regulator in mediating AFB_1_-induced neurotoxicity. This study exemplifies one approach for using clinically developed synthetic or natural sEH inhibitors to reduce the toxicity induced by chronic AFB_1_ exposure in humans. Further preclinical and clinical studies include the following: the efficacy, drug dose optimization, brain penetration, neuro pharmacokinetics, and safety characteristics of sEH inhibitors in the brain for alleviating neurotoxicity in multiple models of AFB_1_ exposure. In addition, this study suggests that future studies should also investigate the effects of targeting sEH in alleviating other foodborne or environmental toxins-associated neuroinflammation and dopaminergic neurotoxicity.

## 4. Materials and Methods

### 4.1. Animal Study

All animal experiments were performed in accordance with the protocol approved by the Institutional Animal Care and Use Committee (IACUC, protocol # 21628) of the University of California-Davis. All the mice were maintained on a standard chow diet *ad libitum* and great care was taken to ensure the welfare of the included animals. 

#### 4.1.1. Animal Experiment 1: Effects of AFB_1_ Exposure on Neuroinflammation and Neurotoxicity in Mice

C57BL/6J male mice (8-week-old) were purchased from Charles River and randomly assigned to two groups (AFB_1_ or vehicle-treated group, *n* = 6 mice per group). AFB_1_ (Sigma-Aldrich, St. Louis, MO, USA, catalog # A6636) was firstly dissolved in dimethyl sulfoxide (DMSO, Thermo Fisher Scientific, Hampton, NH, USA) and then added into sterile drinking water at a final concentration of 1500 ppb (final concentration of DMSO is 0.1%). The AFB_1_ drinking water was freshly prepared every other day and put in aluminum foil-wrapped bottles to reduce the photochemical degradation of AFB_1_ and minimize degradation-caused dose changes. After 21 days, the mice were euthanized using isoflurane (5%, Dechra Pharmaceuticals, Northwich, UK) inhalation followed by cervical dislocation. The frontal brain regions (striatum and an associated portion of the cerebral cortex) were dissected for further analysis.

#### 4.1.2. Animal Experiment 2: Effects of sEH Deficiency or Inhibition on AFB_1_-Induced Neuroinflammation and Neurotoxicity in Mice

C57BL/6J male WT mice or *Ephx2*^−/−^ mice (8-week-old, *n* = 8 mice per group, maintained at the University of California, Davis) were treated with AFB_1_ (1500 ppb) or the vehicle (drinking water containing 0.15% DMSO). To determine the effect of pharmacological inhibition of sEH, another group of C57BL/6J male mice WT mice (8-week-old) were given AFB_1_ (1500 ppb) together with the potent sEH inhibitor EC5026 (10 mg/L) in the vehicle (drinking water containing 0.15% DMSO). EC5026 was synthesized as previously described [50]. The vehicle, AFB_1_ only, and AFB_1_ plus EC5026 drinking water were freshly prepared every other day and put in aluminum foil-wrapped bottles. After 21 days, the mice were euthanized using isoflurane (5%) inhalation followed by cervical dislocation. The frontal brain regions (striatum and an associated portion of the cerebral cortex) were dissected for further analysis.

### 4.2. Total RNA Isolation and Quantitative Polymerase Chain Reaction (qPCR) Analysis 

The frontal brain regions were dissected and ground after being frozen in liquid nitrogen. The TRIzol reagent (Ambion, Austin, TX, USA) was used to isolate the total RNA according to the manufacturer’s instructions. A NanoDrop Spectrophotometer (Thermo Fisher Scientific, Waltham, MA, USA) was used to measure the quality and quantity of the extracted RNA. A High-Capacity cDNA Reverse Transcription kit (Applied Biosystems, Waltham, MA, USA) was used to reverse transcribe RNA into cDNA. An Mic qPCR Cycler (Bio Molecular Systems, Upper Coomera, Australia) was used to perform qPCR. The results of target genes were normalized to glyceraldehyde-3-phosphate dehydrogenase (*Gapdh*) and the final gene expressions were calculated using the 2^−ΔΔCt^ method. All the murine primers used in this study were obtained from Thermo Fisher Scientific. The information on these primers is listed in Appendix A. 

### 4.3. Protein Extraction and Immunoblotting

Protein extraction and immunoblotting were performed as described [51]. Proteins from the frontal brain regions were extracted with RIPA lysis buffer (Boston BioProducts, Milford, MA, USA) with a protease inhibitor cocktail (Boston BioProducts). A BCA protein assay kit (Thermo Fisher Scientific) was used to determine the protein concentrations. The tissue samples (15 μg) were resolved on SDS-PAGE gels (Bio-Rad Laboratories, Hercules, CA, USA) and transferred onto nitrocellulose membranes (Bio-Rad Laboratories). The membrane was blocked in 5% milk buffer, incubated with primary antibodies against Iba-1 (Cell Signaling, Danvers, MA, USA, catalog # 17198), IκBα (Cell Signaling, catalog # 4814), phospho-α-synuclein (Ser129) (Cell Signaling, catalog # 23706), α-synuclein (Cell Signaling, catalog # 4179), and tyrosine hydroxylase (EMD Millipore, Burlington, MA, USA, catalog # AB152) in 5% BSA solution at 4 °C overnight. The membranes were incubated with HRP-linked anti-rabbit, or anti-mouse antibodies (Cell Signaling, catalog #7074, #7076). Clarity Western Enhanced Chemiluminescence (ECL) Substrates (Bio-Rad Laboratories) were used for imaging. The chemiluminescence was developed by Clarity Western Enhanced Chemiluminescence (ECL) Substrates (Bio-Rad Laboratories) and detected using Bio-Rad ChemiDoc™ Imaging Systems. Quantification of immunoblotting was performed using Image J software (Version 1.53v). β-actin (Sigma-Aldrich, catalog # A2228) was used as a loading control and for normalization.

### 4.4. Tissue Staining 

The frontal brain tissues were fixed in 4% paraformaldehyde overnight at 4 °C, transferred to 30% sucrose solution for dehydration, and sliced into 12 μm sections using the cryostat (Leica Biosystems, Wetzlar, Germany). Antigen retrieval was performed by heating the sections in 0.01 M citrate buffer (pH 6.0) to 95 °C for 5 min. Immunohistochemistry staining was conducted using the HRP/DAB (ABC) Detection IHC kit (Abcam, Cambridge, UK) according to the manufacturer’s instructions. The anti-Iba-1 (Cell Signaling, catalog # 17198) was used to probe the target protein in the tissue section. The number of Iba-1^+^ cells per field was counted using Image J software (Version 1.53v).

### 4.5. Enzyme-Linked Immunosorbent Assay (ELISA) Analysis of Dopamine 

Proteins from the frontal brain regions were extracted using PBS with sonication on ice and kept in −80 °C. Protein concentrations were determined using the BCA protein assay kit. The level of dopamine in brain lysis was determined using the mouse dopamine ELISA kit (MyBiosource, San Diego, CA, USA, catalog # MBS162171) according to the manufacturer’s instructions. The dopamine level from each sample was normalized using the protein concentration of the corresponding sample and was expressed in a unit of ng/mg of tissue protein.

### 4.6. sEH Activity Measurement

The frontal brain regions were lysed to measure the sEH activity. sEH activity was measured as previously described [52]. Briefly, to 100 µL of tissue suspension, 1 µL of a 5 mM solution of [^3^H]-trans-diphenyl-propene oxide (t-DPPO) in DMSO was added ([S]_final_ = 50 µM; 10,000 cpm) and was incubated at 37 °C for 30 min. The reaction was quenched by the addition of 60 µL of methanol and 200 µL of iso-octane. The enzymatic activity was determined by measuring the quantity of radioactive diol formed in the aqueous phase using a scintillation counter (TriCarb 2810 TR, Perkin Elmer, Shelton, CT, USA). The enzymatic activity sample was normalized using the corresponding protein concentration.

### 4.7. Cell Assays

The N27 rat dopaminergic neural cells were obtained from EMD Millipore (catalog # SCC048). N27 cells were cultured in RPMI 1640 (Corning, New York, NY, USA, catalog # 10040CVa) supplemented with 10% fetal bovine serum. All cells were maintained in an atmosphere of 5% CO_2_ and at 37 °C. 

For the MTT assay, N27 cells were seeded in 96-well plates at a density of 30,000 cells/well and incubated for 24 h. Cells were then treated with AFB_1_ (0.1, 1, 10 μM) or the vehicle control (DMSO, 0.1%) for 24 h. After that, cells were washed with PBS, 0.5 mg/mL MTT (catalog # M6494, Thermo Fisher Scientific) in fresh RPMI 1640 was added to each well and then incubated for 2 h at 37 °C. After the supernatant was removed, 100 mL of DMSO was added to each well. The difference in absorbance at 570 nm was measured on a microplate reader (Molecular Devices). The results were expressed as percentages of the control (%).

For quantitative PCR analysis, N27 cells were seeded in 12-well plates at a density of 300,000 cells/well and incubated for 24 h. Cells were then treated with AFB_1_ (10 μM) with or without EC5026 (0.5 μM) for 24 h. Total RNA extraction, reverse transcription, quantitative PCR, and data analysis were performed as described above. The information on rat primers is listed in Appendix A. 

### 4.8. Statistical Analysis

All data are expressed as the mean ± standard error of the mean (SEM). The Shapiro–Wilk test was used to verify the normality of data and Levene’s mean test was used to assess the equal variance of data before the statistical analysis. Statistical comparison of the two groups was performed using either Student’s *t*-test or the Wilcoxon–Mann–Whitney test (when the normality test failed). Statistical comparison of three groups was determined using one-way ANOVA followed by Tukey’s or Fisher’s post hoc test, or using the Kruskal–Wallis test on ranks (when the normality test failed), followed by the appropriate post hoc test. All the data analyses were performed by using SigmaPlot software (Version 11, Systat Software, Inc., Chicago, IL, USA). *p* values less than 0.05 were reported as statistically significant.

## Figures and Tables

**Figure 1 ijms-24-09938-f001:**
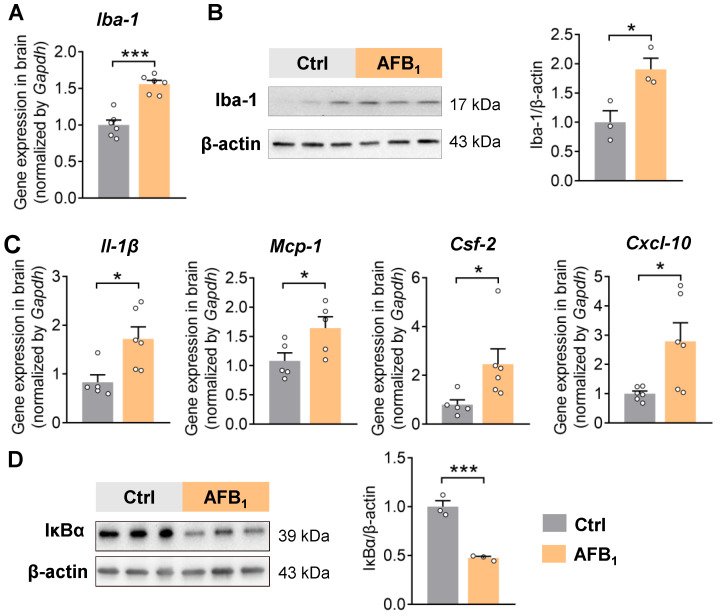
AFB_1_ increases neuroinflammation. (**A**) Gene expression of microglia marker *Iba-1* in the brain. (**B**) Immunoblotting analysis of Iba-1 in the brain (*n* = 3 mice per group). (**C**) Gene expression of pro-inflammatory cytokines *Il-1β*, *Mcp-1*, *Csf-2*, *and Cxcl-10* in the brain. (**D**) Immunoblotting analysis of IκBα in the brain (*n* = 3 mice per group). The results are expressed as mean ± SEM. *n* = 5–6 mice per group. The statistical significance of the two groups was determined using Student’s *t*-test or Wilcoxon–Mann–Whitney test. * *p* < 0.05, *** *p* < 0.001.

**Figure 2 ijms-24-09938-f002:**
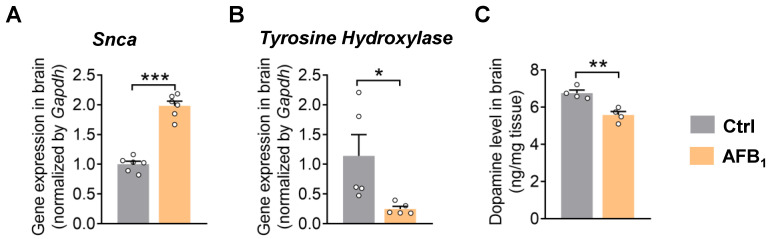
AFB_1_ stimulates α-synuclein upregulation and causes dopaminergic neuron dysfunction. (**A**) Gene expression of *Snca* in the brain. (**B**) Gene expression of dopaminergic neuron marker *tyrosine hydroxylase* in the brain. (**C**) Level of dopamine in the brain. The results are expressed as mean ± SEM. *n* = 4–6 mice per group. The statistical significance of the two groups was determined using Student’s *t*-test or Wilcoxon–Mann–Whitney test. * *p* < 0.05, ** *p* < 0.01, *** *p* < 0.001.

**Figure 3 ijms-24-09938-f003:**
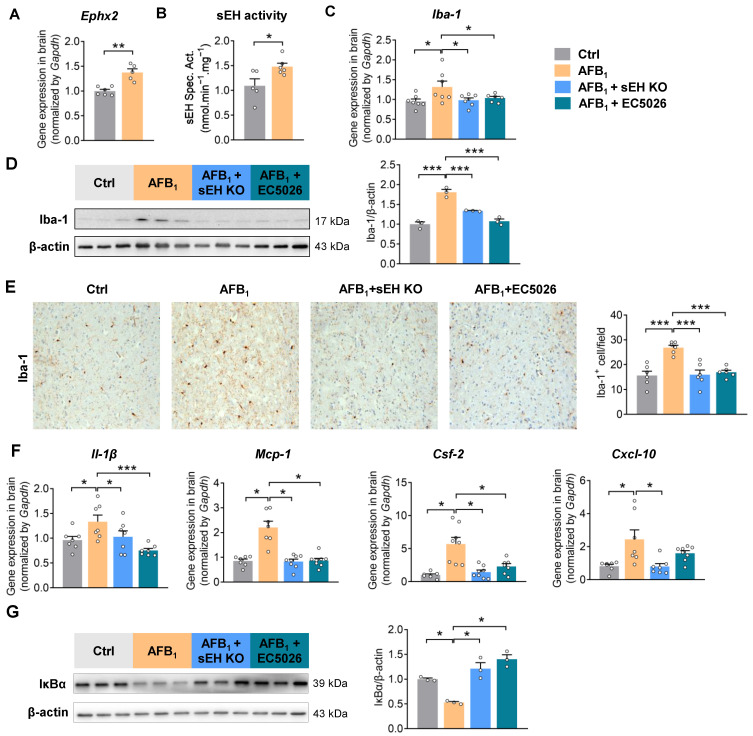
sEH deficiency or inhibition attenuates the AFB_1_-induced neuroinflammation. (**A**) Gene expression of *Ephx2* in the brain. (**B**) Enzymatic activity of sEH in the brain. (**C**) Gene expression of microglia marker *Iba-1* in the brain. (**D**) Immunoblotting analysis of Iba-1 in the brain. (**E**) Immunohistochemical staining and quantification of Iba-1^+^ (magnification 200×) in the brain. (**F**) Gene expression of pro-inflammatory cytokines *Il-1β*, *Mcp-1*, *Csf-2*, and *Cxcl-10* in the brain. (**G**) Immunoblotting analysis of IκBα in the brain (*n* = 3 mice per group). The results are expressed as mean ± SEM. *n* = 7–8 mice per group. Statistical significance was determined using one-way ANOVA or Kruskal–Wallis test on ranks. * *p* < 0.05, ** *p* < 0.01, *** *p* < 0.001.

**Figure 4 ijms-24-09938-f004:**
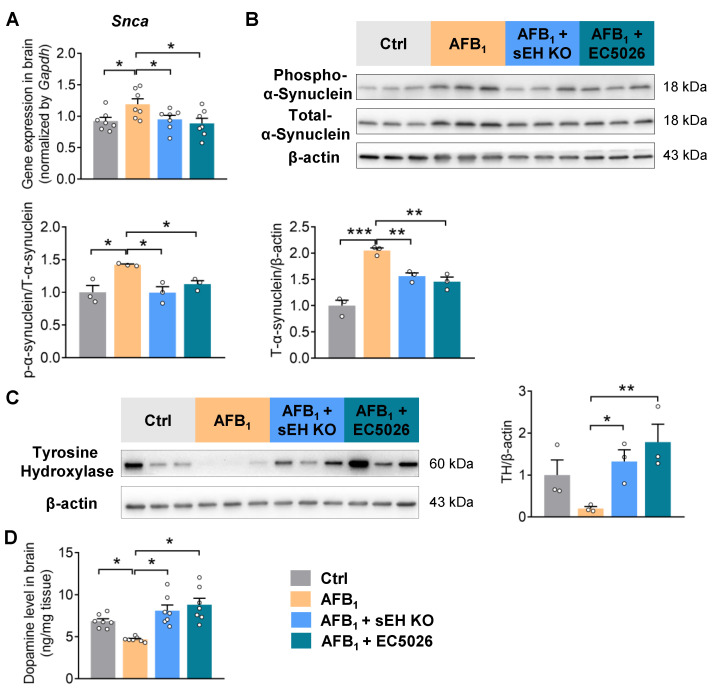
sEH deficiency or inhibition alleviates the AFB_1_-stimulated α-synuclein upregulation and dopaminergic neuron dysfunction. (**A**) Gene expression of *Snca* in the brain. (**B**) Immunoblotting analysis of total and phosphorylation of α-synuclein in the brain (*n* = 3 mice per group). (**C**) Immunoblotting analysis of dopaminergic neuron marker tyrosine hydroxylase in the brain (*n* = 3 mice per group). (**D**) Level of dopamine in the brain. The results are expressed as mean ± SEM. *n* = 7–8 mice per group. Statistical significance was determined using one-way ANOVA or Kruskal–Wallis test on ranks. * *p* < 0.05, ** *p* < 0.01, *** *p* < 0.001.

## Data Availability

All data are available in the main text or the Appendix A. Additional raw data are available on request.

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
