# Peer review of "Aflatoxin B1 Increases Soluble Epoxide Hydrolase in the Brain and Induces Neuroinflammation and Dopaminergic Neurotoxicity"

_ijms, 2023, doi:10.3390/ijms24129938_

Round 1
Reviewer 1 Report
1. Could the authors revise references for less than 10% of auto citations? (nine for Bruce D. Hammock)
Others:
a. Could the authors provide references in the journal’s style?
Reviewer 2 Report
Please add why only male animals are used. Is that to reflect the male dominance in PD n humans or what? This is important because the ARRIVE guidelines recommend talking about both animal sexes in the experiments so that we can see f a sex-related difference might be evident in any responses. The information then might be useful related to human care at clinics, and sex-related knowledge in this regard.
The authors have presented the clinical relevance of their data, but very short and superficial. Please elaborate further on this sentence "This 241 study highlights the future direction for developing novel approaches to reducing the toxic 242 effects of chronic AFB1 exposure in humans."
Please add the major limitations of your study.
Please evaluate the internal and external validity of the findings. i.e. can the findings be generalized or further studies are required to make a sharp conclusion?
Reviewer 3 Report
The authors administered aflatoxin B1 (AFB1) to male mice in drinking water for 21 days to induce neurotoxicity. They found that AFB1 increases gene expression of pro-inflammatory cytokines and also treatment upregulates a-synuclein and tyrosine hydroxylase gene expression in the brain leading neurotoxicity. In addition, AFB1 increases the expression of soluble epoxide hydrolase (sEH), related to several diseases, so the authors explore the effects of sEH deficiency or inhibition in AFB1-induced neurotoxicity. The results indicate that this approach it could be a promising target to prevent neurotoxicity caused by AFB1. The experiments were well-designed, and the results support the conclusions. However, there are some points mentioned below that should be corrected.
Minor points:
1. Authors should explain why the number of animals per group is 3 to 6, and why the Figure of Iba-1 protein expression of Iba-1 there are only 3 animals whereas in gene expression there are 6 animals per group.
2. On the representative western blot images, add the molecular weight of each protein.
3. Please in Table S1, indicate the annealing temperature for each primer.
4. Line 194. The word “in vitro” must be written in italics.
5. The authors could complete the discussion section on the safety of this approach or suggest what studies are needed to demonstrate the safety and effectiveness of this therapeutic approach.
